# The perception and misperception of optical defocus, shading, and shape

Scott WJ Mooney*, Phillip J Marlow, Barton L Anderson

School of Psychology, The University of Sydney, Sydney, Australia

**Abstract** The human visual system is tasked with recovering the different physical sources of optical structure that generate our retinal images. Separate research has focused on understanding how the visual system estimates (a) environmental sources of image structure and (b) blur induced by the eye's limited focal range, but little is known about how the visual system distinguishes environmental sources from optical defocus. Here, we present evidence that this is a fundamental perceptual problem and provide insights into how and when the visual system succeeds and fails in solving it. We show that fully focused surface shading can be misperceived as defocused and that optical blur can be misattributed to the material properties and shape of surfaces. We further reveal how these misperceptions depend on the relationship between shading gradients and sharp contours, and conclude that computations of blur are inherently linked to computations of surface shape, material, and illumination.

DOI: https://doi.org/10.7554/eLife.48214.001

*For correspondence:
scm2011@med.cornell.edu

## Introduction

An image of a surface is a product of its three-dimensional (3D) shape, the reflectance and transmittance behavior of its material, the surrounding light sources (the 'light field'), and the focal parameters of the imaging lens. All of these sources are conflated in the light that reaches the eyes, yet we nevertheless perceive distinct impressions of shape, material, illumination, and focus. One of the fundamental goals of mid-level vision research is to understand how the visual system extracts these different sources of structure.

Most research into this problem has focused on how the visual system extracts environmental sources of structure – 3D shape, reflectance (color, lightness, gloss), and surface opacity (transparency, translucency, and subsurface scattering). But the focal properties of single-chambered eyes also contribute to the optical structure projected to the retinae. Eyes that utilize refraction to generate a focused image are subject to depth of field defocus, which causes some image regions to be blurred. Optical defects can also cause images to be globally blurred for the increasing number of people that require optical correction. Although there has been a significant body of research into cues that affect the severity of perceived blur (*Ciuffreda et al., 2006*; *Crete et al., 2007*; *Ferzli and Karam, 2006*; *Pentland, 1987*; *Tadmor and Tolhurst, 1994*; *Webster et al., 2002*) and the role of depth of field defocus as a cue to depth (*Held et al., 2010*; *Marshall et al., 1996*; *Mather, 1996*; *Mather, 1997*; *Mather and Smith, 2000*; *Mather and Smith, 2002*; *O'Shea et al., 1997*; *Watt et al., 2005*), it is still unknown how the visual system distinguishes blur from environmental sources of low spatial frequency image structure.

The computational problem of discriminating optical defocus from environmental sources of low frequency structure does not appear to have been explicitly addressed previously. This may be due to the absence of empirical evidence that the visual system can misattribute image gradients produced by environmental sources to defocus or misattribute gradients produced by defocus to environmental sources. Here, we provide evidence of both. We show that defocus can be experienced in

**eLife digest** We perceive the visual world as made of objects of different shapes, sizes and colors. Some may be smooth, shiny and reflective, whereas others are rough and uneven; some may be in shadow, while others are brightly lit. The brain must identify and distinguish all of these different features to build an accurate, three-dimensional model of the environment.

Information about any visual feature originates as light bouncing off an object and entering the eye, which then captures the reflected light and focuses it onto the retina. There, cells generate electrical signals for the brain to process. However, different types of visual features can result in the same pattern of activity. The brain must rely on prior knowledge and educated guesses to disentangle the contributions made by different features, but we know little about the processes that make this possible.

Here, Mooney et al. examine how the visual system can tell whether an object is blurry, or if it presents the smooth light-to-dark shading that can accompany curved shapes. The experiments show that images of shaded curved surfaces can appear blurry even when they are fully in focus. However, adding a specific type of sharp edge, called a bounding contour, eliminates this illusion. This suggests that the brain uses these sharp edges to judge whether an image is in focus. In fact, adding bounding contours can trick the visual system into perceiving a blurry image as sharp.

Understanding how the human visual system interprets images could lead to advances in computer vision. Artificial vision systems – such as those used in face or license plate recognition – must determine which parts of an image are in focus before attempting to extract visual information. Identifying the cues that enable the human visual system to solve this problem could help to train computers to do the same.

DOI: https://doi.org/10.7554/eLife.48214.002

fully focused images and that optical defocus can be misperceived as distortions in the perceived 3D shape of smoothly shaded surfaces.

Consider the surface depicted in *Figure 1*, which was created by illuminating a smooth (i.e., differentiable) Lambertian ('matte') surface with a collimated light source. The surface has shallow surface relief to avoid the formation of sharp attached shadows and is viewed along the axis of relief to avoid the formation of self-occluding contours. Although rendered as a fully focused surface, this image elicits a strong perception of blur; while some 3D shape from shading may be perceived, the surface appears 'contaminated' by optical defocus.

The perceptual conflation of low frequency shading gradients and optical focus does not appear to have been previously reported. Most research into the perception of shading has attempted to understand how shading provides information about 3D shape using images where it was assumed or somehow 'known' that the intensity gradients were caused by focused patterns of shading. The potential conflation of shading and blur may have been overlooked because of the particular surface geometries and viewing conditions that were used in these studies. Most previous work on shape from shading has studied images that contained sharp contours generated by either smooth self-occlusions or abrupt bounding contours (e.g. *Horn and Brooks, 1989*; *Koenderink et al., 2001*; *Mingolla and Todd, 1986*; *Pentland, 1984*; *Ramachandran, 1988*; *Todd et al., 1996*), while experiments that have used 'terrain' surfaces similar to *Figure 1* have predominantly been investigations of illumination perception (*Koenderink et al., 2004*; *Koenderink et al., 2007*). This suggests that the visual system may exploit sharp contours in generating percepts of focus when viewing low frequency shading gradients. If so, it should be possible to eliminate the perception of illusory blur in *Figure 1* by introducing sharp bounding contours.

We informally tested this hypothesis by constructing the images depicted in *Figure 2*. The image on the left (2A) was generated by intersecting the surface in *Figure 1* with a gray plane parallel to the direction of its relief and occluding all surface regions beyond the depth of that plane (a particular form of 'planar cut' dubbed a 'level cut'). Informal observation suggests that this manipulation enhances the perceived 3D shape of the surface and completely eliminates the perception of blur. Note also that the contour appears to be unambiguously 'owned' by (attached to) the shading gradients rather than the homogeneous gray regions. Now consider the image in *Figure 2B*, which was

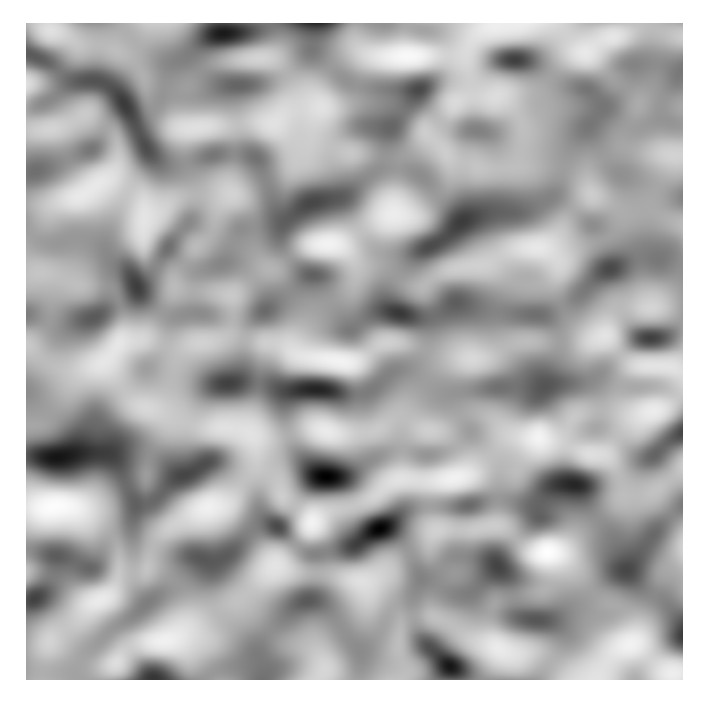

**Figure 1.** A deformed matte terrain illuminated by a light source elevated 45° above the line of sight. Despite being rendered in full focus, the image appears blurry, and the local features of its 3D shape seem difficult to perceive.

DOI: https://doi.org/10.7554/eLife.48214.003

created by treating the gray regions in *Figure 2A* as parts of a single gray 'mask' and rotating it 180 degrees over the shaded surface. This image contains the exact same contours as *Figure 2A*, but the gray regions occlude different portions of the shaded surface. There are a number of striking perceptual differences evoked by comparison of *Figure 2B* and *Figure 2A*. First, whereas the perceived shape from shading is *enhanced* in *Figure 2A*, the perception of shape from shading is *impaired* by the contours in *Figure 2B*. A second difference is that the border ownership of the contour in *Figure 2B* is ambiguous; whereas the contours in *Figure 2A* appear unambiguously attached to the shading gradients, the contours in *Figure 2B* can appear attached to either side. The way the border ownership is perceived can have a dramatic impact on how the gradients are perceived. When the contours appear attached to the shading gradients, the gradients are perceived as unstructured 2D 'noise' (such as variations in pigment) without any clear sense of 3D shape or optical focus. But when the contours appear attached to the gray mask, the shaded surface appears as a partially occluded shaded surface that is just as blurred as the surface in *Figure 1*.

What is responsible for the striking perceptual differences observed in *Figure 2A* and *Figure 2B*? Why does one set of contours eliminate the perceived blur, enhance perceived 3D shape, and unambiguously determine the side of the contour that is figure, while the other does not? We suggest here that the perceived focus and enhanced perception of 3D shape arises because the level cut images approximate the geometric and photometric image properties generated by smooth self-occluding contours. More specifically, we will argue that there are two constraints that play a causal role in eliciting the perception of focus in images of shaded surfaces: photogeometric constraints that arise generically along smooth self-occlusions; and the attachment of shading gradients to convex surfaces. We consider each in turn.

The first constraint arises from the physics of surface reflectance and the projective geometry of smooth self-occluding rims. Both shading intensity and bounding contour shape depend on the same environmental property – the local 3D shape of the surface – and are therefore inherently linked. This link makes it possible to combine two well-known constraints about the shape of the

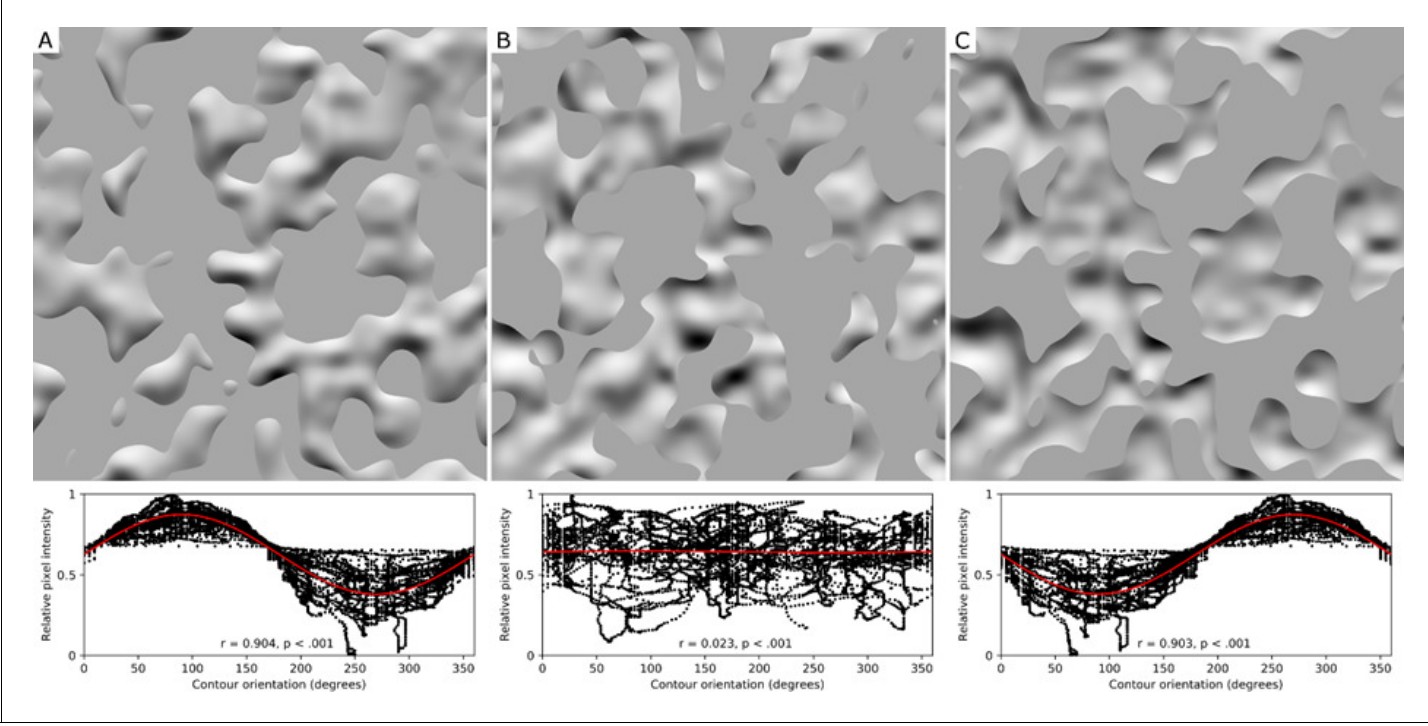

**Figure 2.** Orientation-intensity covariation induced by level cut contours. (**A**) The same shaded surface as shown in *Figure 1*, but now partially occluded by a gray level cut mask. The mask's contours were created by intersecting the deformed terrain with a flat plane, and completely eliminate the percepts of blur experienced in *Figure 1*. The plot beneath reveals how image intensity at the contour varies as an approximate cosine function of orientation. Every pixel of shading along the contour is represented in this plot as a black dot. The correlation coefficient of the cosine fit and its p-value are shown within the graph. (**B**) The level cut mask that occludes the gradients in (**A**) has been rotated by 180 degrees over the image, which eliminates the relationship between the contours and surface geometry. The plot below the image reveals the destructive effect of this rotation on the systematic covariation between contour orientation and image intensity. Here, the shading gradients are misperceived as either a flat texture or a blurred surface beneath a floating stencil. (**C**) The contours shown in (**A**) now form a mask that occludes every part of the shaded surface in front of the level cut. The orientation-intensity covariation along the contour is equally strong, but the 180° shift in its phase causes the gradients to appear bistable: either bumps lit from below, or concavities lit from above. These gradients appear less focused overall than the unambiguously convex surface in (**A**).
DOI: https://doi.org/10.7554/eLife.48214.004

The following figure supplements are available for figure 2:

**Figure supplement 1.** The 'convex' level cut (top) and rotated (bottom) mask conditions used in Experiment 1.
DOI: https://doi.org/10.7554/eLife.48214.005

**Figure supplement 2.** The 'bistable' level cut (top) and rotated (bottom) mask conditions used in Experiment 1.
DOI: https://doi.org/10.7554/eLife.48214.006

contours generated by smooth self-occlusions and the shading intensity into a novel constraint (*Marlow et al., 2019*). First, it is known that the 3D shape along a smooth self-occluding rim can be derived from its 2D image contour: the local slant of the contour relative to the observer (i.e., how much it deviates from fronto-parallel) is constant (approximately 90 degrees), and the tilt of the surface (the direction in which it slants away from the observer) is specified by the orientation of the rim's image contour (*Barrow and Tenenbaum, 1978*). Second, it is known that local 3D surface orientation is primarily responsible for shading intensity (*Horn and Brooks, 1989*), subject to some modification by vignetting and/or interreflections (*Langer and Zucker, 1994*). The mutual dependence of contour orientation and shading intensity on the same local 3D surface orientation causes shading to covary with the orientation of the rim's projected image contour: for a Lambertian surface illuminated by a collimated light source, intensity will decline as a cosine function of contour orientation relative to the brightest point along the contour (i.e., the orientation most closely aligned with the illumination direction). This relationship holds exactly for surfaces illuminated by collimated light sources (see Materials and methods), and we have previously shown that it is statistically robust in

natural light fields that contain multiple sources of illumination (*Marlow et al., 2019*). We refer to this relationship as the *orientation-intensity covariation* of the contour and its adjacent shading.

The covariation of contour orientation and intensity along smooth self-occluding rims generalizes to other types of bounding contours that have been shown to affect perceived shape from shading, such as 'planar cuts' (i.e., contours formed by 'slicing' a shaded surface with a plane; *Marlow et al., 2019*). The level cut image in *Figure 2A* is one example of a planar cut. The orientation-intensity covariation that arises along planar cuts of shaded surfaces is similar to that generated by smooth self-occluding rims (see Materials and methods). The relationship between contour orientation and shading intensity along the contours in *Figure 2* is depicted in the plot below each image. Note that *Figure 2A* exhibits a clear cosine-like relationship, whereas *Figure 2B* exhibits no covariation at all. However, this covariation is generally weaker for planar cuts than for self-occluding rims: more than one 3D surface orientation can generate the same 2D contour orientation, which means that identically oriented planar cut segments can project different shading intensities in the image (as can be seen upon close examination of *Figure 2A*). Nonetheless, we previously showed that planar cuts exhibit a robust orientation-intensity covariation (apart from a few degenerate cases; *Marlow et al., 2019*), which suggests that this covariation could provide a reliable cue that the visual system uses to identify the bounding contours of shaded surfaces.

The second property that links the level cut image in *Figure 2A* to images containing smooth self-occlusions is that they are both globally convex (*Koenderink, 1984*). If the covariation of intensity and contour orientation along sharp bounding contours is sufficient to explain the perception of focus in shaded surfaces, then it should not matter whether these contours bound a convex or concave surface. The importance of convexity can be assessed by constructing a level cut surface that removes all of the convex surface regions that appear in front of the cut and displaying only the 'valleys' (*Figure 2C*). This stimulus exhibits the same orientation-intensity covariation, but is inherently ambiguous: it can be perceived as a convex surface illuminated from below or as a concave surface illuminated from above (e.g. *Ramachandran, 1988*; see *Liu and Todd, 2004*). There are two well-established biases that determine how such ambiguities are resolved: a bias to perceive the illumination as coming from above (*Belhumeur et al., 1999*; *Koenderink et al., 2001*) and a bias to perceive surfaces as convex (*Hill and Bruce, 1994*; *Langer and Bülthoff, 2001*). These two biases are aligned in *Figure 2a*, which is presumably why this surface is perceived as a stable convex surface illuminated from above. However, these biases are in conflict in *Figure 2C*, causing some perceptual bistability. Informal observations suggest that perceived focus depends on the perceived convexity or concavity of the surface. When the surface appears convex, no clear percept of blur is experienced; but when it appears concave, the surface appears blurred. This implies that the strength of the orientation-intensity covariation along the contours is not the sole determinant of the perception of shading, focus, and contour attachment; the convexity of the surface is also critical.

Our previous work showed that photogeometric cues along smooth self-occlusions and planar cuts of 1D luminance profiles provide information that predicts when identical luminance gradients are perceived as surface shading of 3D surfaces. The experiments described below were designed to psychophysically assess the relationship between bounding contour orientation, shading intensity, and contour sharpness on the perception of surface shading, optical defocus, and border ownership.

## Results

The goal of Experiment 1 was to test if sharp image contours cause bounded shading gradients to appear more focused when contour orientation covaries with the intensity of the bounded gradients. This covariation is typically exhibited along contours generated by self-occlusions. Self-occlusions feature prominently in prior work on shape from shading, but their contours are difficult to manipulate without altering the geometry of the entire shaded surface (and hence the shading gradients). However, it is possible to approximate the photogeometric behavior along smooth self-occlusions with planar cuts. This was accomplished by slicing the bumpy plane depicted in *Figure 1* with a plane oriented perpendicular to the axis of surface relief (a 'level cut'; see Materials and methods for details). The resulting level cut contours exhibit an orientation-intensity covariation similar to the covariation along self-occluding contours, but unlike self-occlusions, level cuts can be generated anywhere on the surface and place no constraints on local surface curvature. The photogeometric constraints of self-occlusions and level cuts are described in further detail in the Materials and methods.

To assess the role of the photogeometric covariation along bounding contours in the perception of 3D shape, border ownership, and optical focus, three sets of level cut contours were created by intersecting the surface in *Figure 1* with a fronto-parallel plane at different depths along the axis of surface relief. Each set of contours was used to generate four homogenous gray masks. In the first condition, each mask occluded all surface regions that lay at a greater depth than the level cut, leaving only the shaded peaks visible (e.g. *Figure 2A*). The visual system's biases toward interpreting shaded surfaces as convex and top-lit are aligned in this image, and the visible surface regions consequently appear unambiguously convex; we therefore refer to this condition as the 'convex' level cut condition. In a second condition, the same contours were used to generate a complementary mask that removed all regions in *front* of the level cut, leaving only the shaded valleys visible (e.g. *Figure 2C*). The visual system's convexity and illumination biases conflict in this image; the shaded regions may appear as concave dents illuminated from above or convex bumps illuminated from below. This conflict can result in some bistability in the perceived illumination direction and convexity/concavity, so we refer to this as the 'bistable' level cut condition. The masks in the two 'rotated' conditions were created by rotating the level cut masks in the convex and bistable conditions (respectively) by 180° over the underlying gradients, which breaks the orientation-intensity covariation that occurs along the level cuts (e.g. *Figure 2B*). The depth of the intersecting plane used to define the contours determined the relative proportion of visible gradients in each masked image; the three depth values were chosen to create masks that preserved 25%, 50%, and 75% of the gradients. Note that the masks in the 'convex' conditions with 25% gradients visible are the complements of the masks in the 'bistable' conditions with 75% gradients visible, and vice versa. The luminance of each gray mask was set to the average of the original gradient image. The full set of twelve stimuli can be seen in *Figure 2—figure supplement 1* and *Figure 2—figure supplement 2*.

If the visual system uses the orientation-intensity covariation between contours and shading to distinguish attached bounding contours from arbitrary edges, then the level cut contour conditions should induce stronger percepts of focus than the rotated contours, which have no relationship with the shading gradients. However, if the visual system is better at inferring contour attachment when the adjacent surface appears convex, then the covariation alone (which provides no curvature cues) may not fully eliminate the perception of blur. If this is true, then the contours in the 'bistable' level cut condition should produce weaker percepts of focus than the 'convex' level cut condition, as the bistable stimuli may sometimes appear concave.

Observers judged perceived focus in a paired comparison task and judged perceived surface curvature (convex, concave, or neither) in a separate three-alternative classification task (N = 20). The results confirm our hypotheses and informal observations of the stimuli (*Figure 3*) and were analyzed with an ANOVA and appropriate two-sided contrasts. Observers perceived the level cut stimuli (solid lines in *Figure 3A*) as appearing more focused than the rotated stimuli (dotted lines in *Figure 3A*), $F(1, 19) = 64.68, p<0.001$, 95% CI [25.33, 43.15], Cohen's $d = 1.85$, and further perceived the 'convex' level cut stimuli as more focused than the 'bistable' level cut stimuli, $t(19) = 7.05, p<0.001$, 95% CI [20.45, 37.73], Cohen's $d = 1.62$. All observers perceived the 'convex' level cut surfaces as convex bumps on all trials (first plot in *Figure 3B*), whereas the gradients occluded by rotated masks were most likely to be perceived as neither bumps nor dents (third and fourth plots in *Figure 3B*). Perceived focus decreased as a function of increasing gradient visibility in the 'bistable' level cut condition, $t(19) = 7.07, p<0.001$, and the rotated conditions, $t(19) = 5.05, p<0.001$, but not the convex level cut condition, $t(19) = 1.73$, $p = 0.100$, which suggests that the convex level cut masks were the only stimuli that were perceived as (equally) fully focused. In the other three mask conditions, it is likely that observers simply preferred to select images in which less of the perceptually blurred gradients were visible.

The data suggest that the effects of the different contours on perceived focus are mediated by the perception of convexity, which is strongest when the contours exhibit an orientation-intensity covariation with the shading and the convex interpretation is consistent with percepts of top-down illumination. Notably, perceived focus and perceived convexity simultaneously decreased in the 'bistable' conditions – but not the 'convex' conditions – as the ratio of visible gradients increased. There was a significant correlation between perceived focus and the proportion of observers who rated each stimulus as appearing convex in shape, $\rho = 0.904$, $p<.001$. This correlation remained significant even when the unambiguous 'convex' mask conditions were excluded from the analysis, $\rho = 0.777, p = 0.014$. This finding supports our informal observation from *Figure 2*: contours

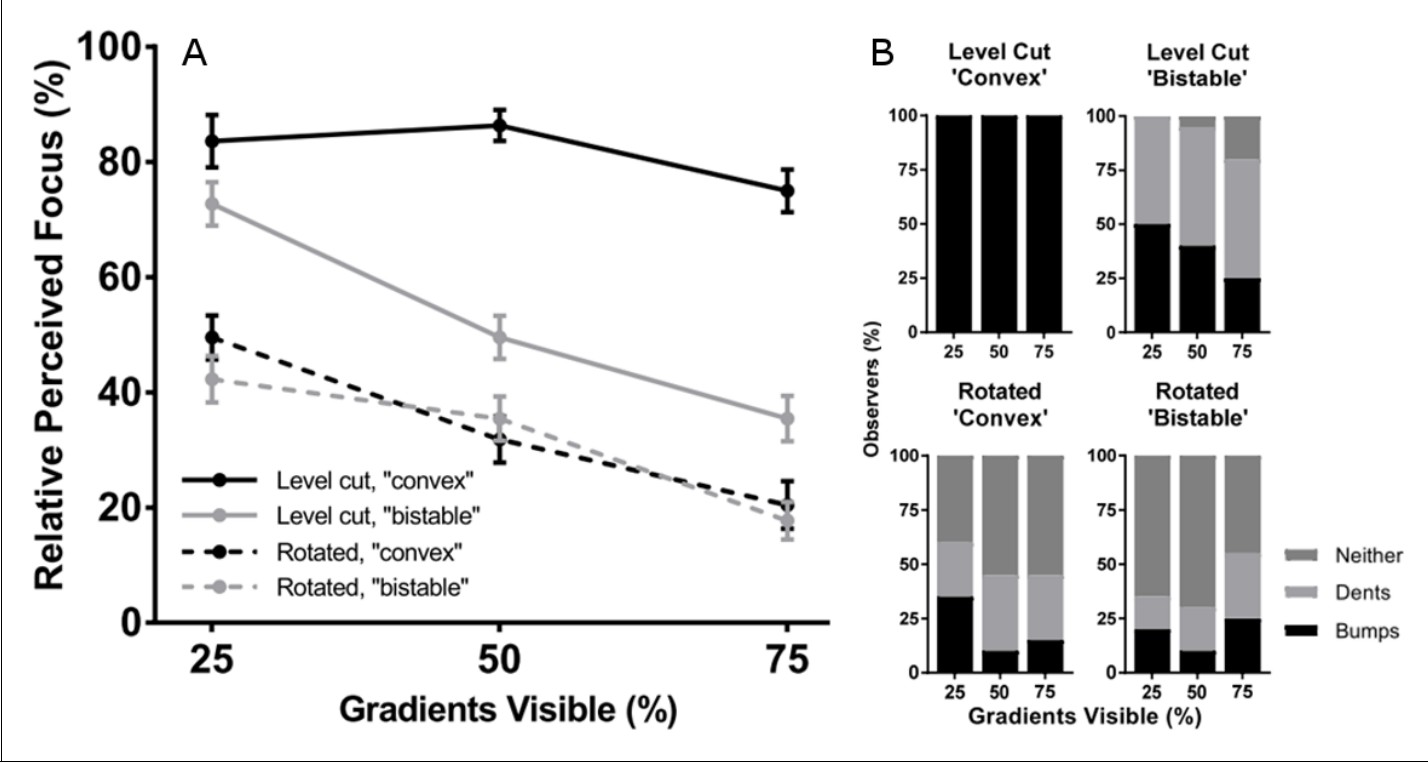

**Figure 3.** Perceived focus and shape in Experiment 1. (A) Perceived focus in Experiment 1. The horizontal axis represents the percentage of gradients visible and the four lines represent the four combinations of mask rotation (level cut vs. rotated) and mask occlusion style ('convex' vs. 'bistable'). The vertical axis represents the percentage of trials in which each stimulus was chosen as appearing most focused. Error bars represent ± 1 S.E.M. (B) Perceived curvature type in Experiment 1. In each stacked column plot, the horizontal axis represents the percentage of visible gradients and the vertical axis represents the percentages of observers who perceived that stimulus as convex bumps (black), concave dents (light gray), or neither (dark gray). Each plot depicts a different combination of mask rotation (level cut vs. rotated) and mask occlusion style ('convex' vs. 'bistable'). The observers were the same group as in (A).

DOI: https://doi.org/10.7554/eLife.48214.007

generate stronger cues to contour attachment when they bound surfaces regions that appear convex, and these contours are therefore more likely to propagate focus cues (produced by their sharpness) to the shaded surface. This may also explain why prior studies using globally convex shaded 3D shapes have not observed any illusory gradient blur: the self-occluding contours of these stimuli not only exhibit a strong orientation-intensity covariation (*Marlow et al., 2019*) but also necessarily bound surface regions that have convex curvature in at least one direction. The bounding contours of surface regions that appear concave, however, can appear as occluding edges of the gray mask, similar to the percept that arises from the rotated contours (*Figure 2B*) when the gray regions appear to 'own' the contours and the surface appears occluded.

The results of Experiment 1 suggest that sharp contours that exhibit an orientation-intensity covariation with nearby shading (such as level cuts) provide information about image focus and enhances percepts of 3D shape within the shaded surface. In Experiment 2, we tested the importance of this covariation directly by parametrically varying the strength of the orientation-intensity covariation along contours. This was accomplished with images of smoothly shaded 'ribbons' on a gray background. These images were created by displaying only the shading gradients immediately adjacent to the mask contours used in Experiment 1. Thus, the only source of information about 3D shape is the relationship between the intensity and orientation along the ribbon. Sixteen unique ribbon paths (two pixels wide) were created by generating more smoothly deformed 3D surfaces and intersecting them with planes to produce level cut contours, but the ribbon was manually shaded in MATLAB according to the orientation of its path. This allowed us to create ribbons that exhibited perfect orientation-intensity covariation that could be parametrically decreased to any arbitrary

value. Ribbons were constructed such that intensity decreased in relative intensity from 0.8 to 0.2 as a linear function of its orientation relative to 90˚ (the brightest point). The covariation was then progressively weakened by gradually adding increasing amounts of random low-frequency noise to the ribbon gradients. The gray background had a relative intensity of exactly 0.5.

*Figure 4* depicts three example stimuli. The ribbons in the left panel exhibit a perfect linear orientation-intensity covariation, which induces a vivid percept of surface relief: some of the gray regions are perceived as 'plateaus' raised above the adjacent recessed regions. The Pearson correlation coefficient between shading intensity and contour orientation (relative to 90˚) measured directly from each image is shown in the lower-right corner. In the center panel, the intensity of a different perfectly covarying ribbon has been mixed in equal proportion with low-frequency noise. A moderate degree of covariation remains, but it is not consistent across the image, and the overall impression of stepped relief is substantially weaker. In the right panel, ribbon intensity has been generated entirely by noise; no covariation is present, and no impression of 3D relief is apparent.

Observers (N = 15) were shown randomly-generated ribbon images with thirteen values of noise proportion ranging from 0% (as in the left panel of *Figure 4*) to 100% (as in the right panel of *Figure 4*). These values were not evenly spaced but were instead selected to produce an approximately uniform distribution of correlation coefficients between 1 and 0 when the covariation was measured in each image. Each observer viewed all possible pairs of these thirteen noise values and were instructed to select the image that appeared more three-dimensional in each pair.

The results confirm our informal experience of *Figure 4*: 3D shape percepts monotonically decreased in strength as a function of increasing ribbon noise (left plot in *Figure 5*). This relationship was verified with a linear regression, with the likelihood of being selected as appearing more three-dimensional decreasing by approximately one percentile for every percentile increase in ribbon noise, $b = -1.025$, $R^2 = 0.923, p < 0.001$. Further analysis revealed that this effect was mediated by the amount of orientation-intensity covariation present in the images: mean perceived 3D shape strength also decreased monotonically with the value of the Pearson correlation coefficient when the correlations measured from all presented stimuli were sorted into bins of width 0.1 (right plot in *Figure 5*). These findings do not directly address the issue of perceived defocus, but do support our hypothesis that the photo-geometric behavior occurring at the very edge of covarying contours (such as the level cut contours in *Figure 2A*) is sufficient to generate the vivid impressions of contour attachment observed in the 'convex' conditions of Experiment 1. The data also reinforce our previous findings on the importance of this covariation in generating percepts of 3D shaded shape (*Marlow et al., 2019*).

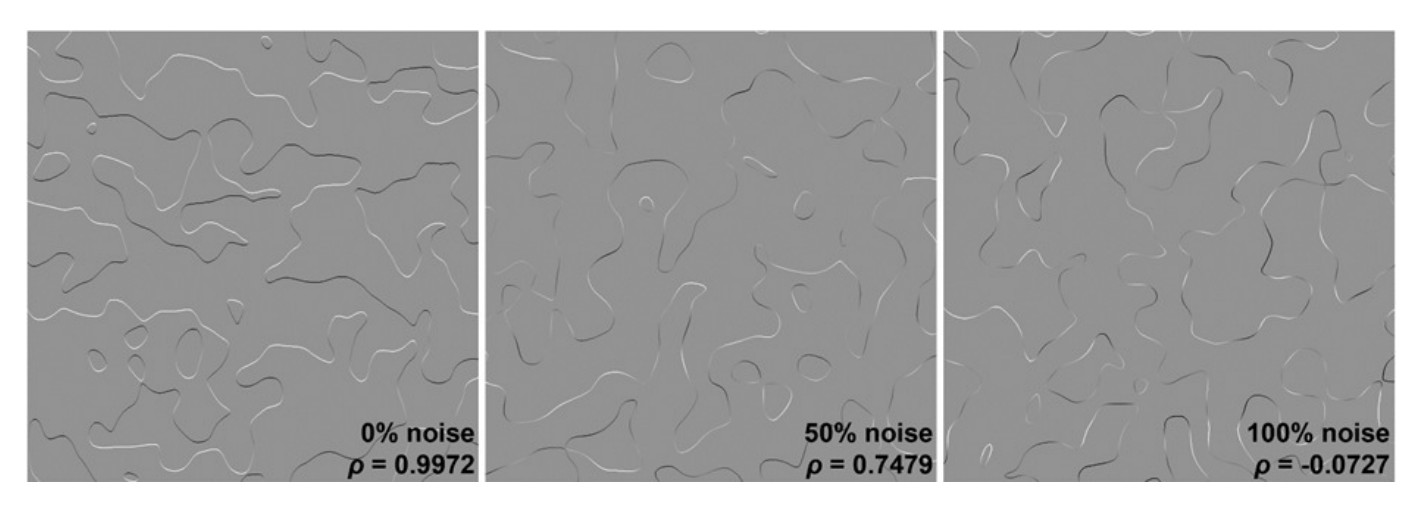

**Figure 4.** Example stimuli used in Experiment 2. The stimuli were created by mixing varying amounts of random noise with shaded ribbons designed to exhibit a perfect linear correlation between orientation (relative to 90˚) and intensity. The examples shown here increase in noise from left to right. The computed global correlations between relative ribbon orientation and intensity are shown in the bottom-right of each stimulus.
DOI: https://doi.org/10.7554/eLife.48214.008

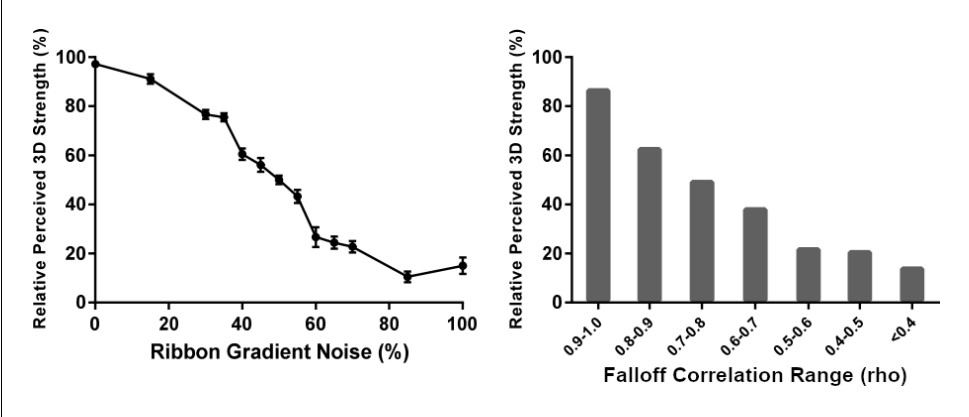

**Figure 5.** Perceived 3D shape in Experiment 2. The horizontal axes represent the percentage of gradient noise in the shaded ribbons (left) and bins of computed Pearson correlation coefficients between ribbon orientation and shading intensity (right). The rightmost bin in the right panel contains all correlation coefficients below 0.4, and all other bins have a width of 0.1. The vertical axis in each plot represents the percentage of trials in which each condition (left) or correlation value (right) was chosen as appearing most vividly 3D out of the total number of trials in which that condition or correlation value appeared. Error bars represent ± 1 S.E.M.

DOI: https://doi.org/10.7554/eLife.48214.009

The results of Experiments 1 and 2 suggest that the *absence* of an orientation-intensity covariation along contours can lead to the misperception of optical defocus and impair the perception of 3D shape. In Experiment 3, we investigated whether image structure that has been optically blurred by defocus can be misperceived as focused if there are sharp contours present nearby that exhibit an orientation-intensity covariation; that is we tested whether the *presence* of such contours can effectively mask the visibility of optically induced blur. We tested this by constructing two variants of the shaded terrain in *Figure 1*. One variant was similar to the level cut image, which contained sharp bounding contours a given relief height (middle of the bottom row of *Figure 6*), and therefore referred to as the level cut image. The other image was constructed by geometrically smoothing the sharp level cut contours of this image, resulting in the image depicted in the middle of the top row of *Figure 6*. As in *Figure 1*, the absence of sharp, intensity-correlated contours causes this surface

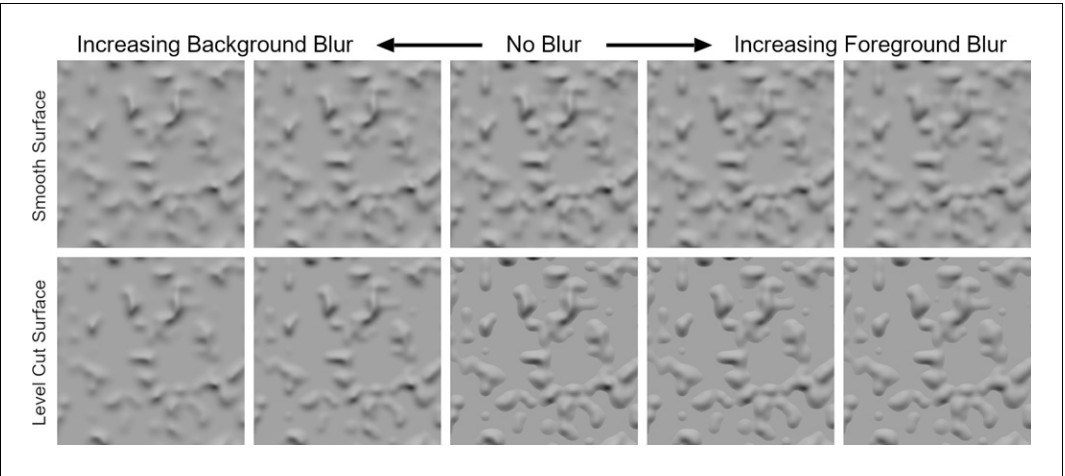

**Figure 6.** Stimuli used in Experiment 3. The top row depicts the 'smoothed' surface with no intersecting plane and the bottom row depicts the 'level cut' surface with the intersecting plane. From left to right, the columns depict the strong background blur, weak background blur, no blur, weak foreground blur, and strong foreground blur conditions.

DOI: https://doi.org/10.7554/eLife.48214.010

to appear blurred even though it was rendered in full focus. We refer to this surface as the 'smoothed' condition. The difference in perceived blur experienced with these two images reinforce the importance of contour sharpness for inducing percepts of focused gradients: the absence of sharp edges in the smoothed condition greatly reduces perceived focus, even though the changes in surface curvature at the edges of the bumps are relatively abrupt.

To assess sensitivity to optical focus, both shaded surfaces were subject to different degrees of optical defocus, with the focal length set to either the background (so the contours surrounding the bumps in the level cut condition remained sharp) or the tips of the bumps (so the contours in the level cut condition were blurred by defocus). The effects of increasing background blur and increasing foreground blur can be seen to the left and right of the center column in *Figure 6*, respectively. The effects of foreground blur on the level cut surface (bottom-right panels) are particularly striking: when the peaks of the surface are affected by defocus blur, the resulting changes in the image gradients do not appear to significantly change the perceived focus of the surface. In the absence of the level cut contours, however, the same defocus manipulation appears to increase apparent blur (top-right panels).

These informal observations were bolstered by psychophysical experiments that measured perceived focus for all ten stimuli using a paired comparison task (N = 10). The results (depicted in *Figure 7*) align with our informal observations of the stimuli in *Figure 6* and were analyzed with appropriate contrasts within an ANOVA (the main effects of which are not relevant here). The vertical axis in *Figure 7* represents the percentage of trials in which each stimulus was selected as appearing more focused, and the horizontal axis represents the different blur conditions. The blue and red lines depict perceived focus for the smoothed and level cut conditions, respectively. The data reveal that defocus blur can be misperceived as shading: the three stimuli with sharp level cut contours (i.e. the level cut condition with no blur or foreground blur) were perceived as the most focused, even when the peaks of the surface were actually blurred by moderate or severe foreground defocus. Statistical analysis revealed that foreground blur had a significantly larger (more negative) effect on perceived focus than background blur for the level cut surface, $t(9) = 20.74, p < 0.001$, 95% CI [45.29, 56.38], Cohen's $d = 6.91$, but not the smoothed surface, $t(9) = 0.31, p = 0.763$, 95% CI [−11.49, 8.71]. Furthermore, the fully-focused smoothed surface with no contours did not significantly differ in perceived focus to the level cut stimulus with weak background blur, $t(9) = 0.71$, $p = 0.494$, 95% CI [−9.28, 4.84]. These results demonstrate that the presence or absence of sharp contours attached to shading gradients can induce both misperceptions of optical focus or defocus (respectively) when they exhibit a systematic intensity-orientation covariation.

The results of the preceding experiments suggest that the most important mediating factor in the perception of focused shading gradients is the presence of sharp bounding contours that exhibit covariation between their orientation and adjacent shading intensity. In Experiment 3, we found that optical blur was overestimated when these sharp, covarying contours were absent from the image. Actual changes in image focus, however, were effectively undetected by observers when the sharpness of nearby covarying contours was preserved. In Experiment 4, we tested whether this underestimation of optical blur occurs because the smooth gradient structure is misattributed to environmental sources: that is whether the optical effects of defocus are attributed to the 3D shape and reflectance properties of the defocused surface. Surfaces with higher microscopic roughness (e.g. glossy plastic or matte materials) will scatter incoming light in more directions and produce smoother image gradients instead of the sharp, detailed specular reflections produced by low-roughness surfaces (e.g. mirrors). This effect of surface roughness on

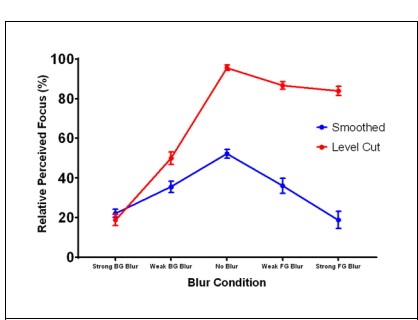

**Figure 7.** Perceived focus in Experiment 3. The horizontal axis represents the five blur conditions and the vertical axis represents the percentage of trials in which each stimulus was chosen as appearing most focused. The blue and red lines depict the results for the smoothed and level cut surfaces, respectively. Error bars represent ± 1 S.E.M.

DOI: https://doi.org/10.7554/eLife.48214.011

image structure is further mediated by curvature: high-curvature surfaces vary in surface orientation more rapidly than low-curvature surfaces, and the shading or reflections they produce are therefore more compressed (i.e., have higher spatial frequency) in the image. We have previously demonstrated that the visual system can misperceive low-curvature, low-roughness surfaces as high-curvature, high-roughness surfaces (*Mooney and Anderson, 2014*), as both combinations of surface properties generate similar gradient structure. The local effects of optical blur on image gradients are similar to the effects of increasing surface roughness or decreasing surface curvature: all three of these physical transformations typically reduce the sharpness of image gradients exhibited by the surface. It is therefore likely that the visual system will have difficulty distinguishing these optical and environmental influences on gradient appearance when they do not produce simultaneous changes in contour sharpness or covariation strength.

We tested this hypothesis by creating three identically shaped surfaces with different reflectance properties: 'matte', 'rough gloss', and 'smooth gloss'. All three materials have an identical diffuse shading component, but the two gloss conditions also contain a specular reflectance component. The amount of scattering in this specular component is greater in the 'rough gloss' condition, which consequently produces reflections with less detail than the 'smooth gloss' condition. The surface's 3D shape had higher curvature than the shape used in Experiment 3 to increase the likelihood of generating measurable misperceptions of 3D shape. Level cut contours were created by intersecting the surface with a plane, as in previous experiments, but the matte gray planar surface itself was here included as part of the rendered scene rather than a mask added to the image afterward. The combined surface was rendered in a natural light field with cast shadows, inter-reflections, and chromatic information. This was done to test whether misperceptions of optical blur occur in more realistic viewing conditions, which are particularly important for the appearance of glossy materials (*Fleming et al., 2003*; *Pellacini et al., 2000*).

Each material condition was rendered with five optical defocus conditions, identical to the conditions in Experiment 3: the 'foreground blur' conditions blurred the peaks of the surface but preserved the sharp contours, and the 'background blur' conditions blurred the sharp contours, leaving the peaks unaffected. The fifteen stimuli are depicted in *Figure 8*, which has a similar layout to *Figure 6*. Each row depicts a different material and each column depicts a different focus condition. As background blur increases to the left of the center column, perceived focus appears to decrease for all three materials. As foreground blur increases to the right, the changes in gradient appearance are misattributed to transformations in material (the surface appears more matte) and 3D shape (the surface appears less curved).

We measured perceived image focus, perceived surface gloss, and perceived 3D shape in three distinct tasks. Perceived focus and perceived surface gloss were each measured from the same group of observers (N = 10) in two separate paired comparison tasks. Perceived 3D shape was measured from expert observers (N = 5) for six of the stimuli (outlined in red in *Figure 8*) using a line of twenty 'gauge figure' probes across the prominent ridge on the left side of each image (red dots in central stimulus of *Figure 8*). These probe settings were integrated to form cross-sectional profiles of perceived relief (see *Koenderink et al., 1992*; *Koenderink et al., 2001*). The results accord with our findings in Experiment three and support our informal observations of the stimuli, which are described in separate sections below for each of the three measured properties.

Observer's reports of perceived focus are depicted in *Figure 9* and were analyzed with appropriate ANOVA contrasts, as in Experiment 3. The vertical axis is the percentage of trials in which each image was selected as appearing more focused. The horizontal axis plots the different blur conditions and each colored line represents a different surface reflectance type. The data indicate that background blur (moving from the central 'no blur' condition to the left) significantly reduced perceived focus for all three materials, $t(9) = -39.62, p{<}0.001$, 95% CI $[-71.86, -64.10]$, Cohen's $d = 13.21$. Foreground blur (moving from the 'no blur' condition to the right) also reduced perceived focus, $t(9) = -9.75, p{<}0.001$, 95% CI $[-29.92, -18.65]$, Cohen's $d = 3.25$, but to a significantly lesser extent than background blur, $t(9) = -34.80, p{<}0.001$, 95% CI $[-46.53, -40.85]$, Cohen's $d = 11.60$. The data also reveal interactions between blur type and material: in the background blur conditions, the surface with smooth gloss was perceived as significantly more focused than the surface with rough gloss, $t(9) = 3.58, p{<}0.005$, 95% CI $[3.16, 13.98]$, Cohen's $d = 1.19$, which was in turn perceived as more focused than the matte surface, $t(9) = 4.27, p = 0.002$, 95% CI $[4.03, 13.11]$, Cohen's $d = 1.42$, but there were no significant differences in perceived focus

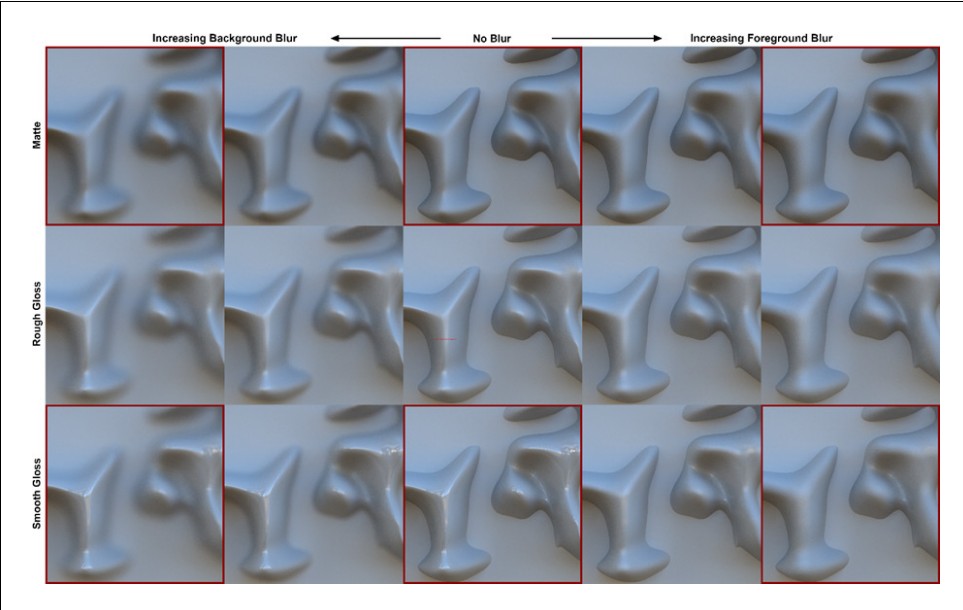

**Figure 8.** Stimuli used in Experiment 4. From top to bottom, the rows depict the surfaces with matte reflectance, rough gloss, and smooth gloss. From left to right, the columns depict the conditions with strong background (BG) blur, weak background blur, no blur, weak foreground (FG) blur, and strong foreground blur. Perceived focus and gloss were measured for all fifteen stimuli. Perceived 3D shape was measured horizontally across the prominent vertical ridge in the six stimuli outlined in red. The probe points where shape measurements were taken are shown as red dots in the central stimulus.

DOI: https://doi.org/10.7554/eLife.48214.012

between materials in the foreground blur conditions, $t(9) = -0.13, p = 0.901$, 95% CI $[-6.67, 5.95]$ (smooth vs. rough gloss) and $t(9) = -1.59, p = 0.146$, 95% CI $[-19.89, 3.47]$ (rough gloss vs. matte). That is, perceived focus decreased as surfaces exhibited more scattering in their reflectance function, but only in the background blur conditions. The effect of material in the background blur conditions suggests that the sharp ridge gradients generated by the more specular materials may have provided useful cues to the focus of the surface peaks. Most of these cues would have been destroyed by defocus in the foreground blur conditions, which may explain why the effect of material disappeared.

Perceived gloss is depicted in *Figure 10* and was analyzed with analogous contrasts to

**Figure 9.** Perceived focus in Experiment 4. The horizontal axis represents the five blur conditions and each colored line represents a different reflectance condition. The vertical axis represents the percentage of trials in which each stimulus was selected as appearing most focused. Error bars represent ± 1 S.E. M.

DOI: https://doi.org/10.7554/eLife.48214.013

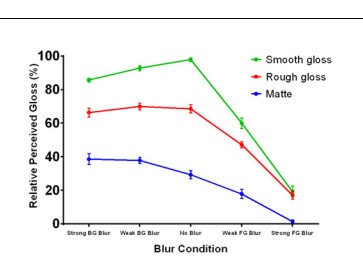

**Figure 10.** Perceived gloss in Experiment 4. The horizontal axis represents the five blur conditions and each colored line represents a different reflectance condition. The vertical axis represents the percentage of trials in which each stimulus was selected as appearing most glossy. Error bars represent ± 1 S.E.M.

DOI: https://doi.org/10.7554/eLife.48214.014

perceived focus. The layout of the plot is identical to *Figure 9*, but the vertical axis now represents the percentage of trials in which each image was selected as appearing glossier. The data indicate that foreground blur (right side) had a large negative effect on perceived gloss relative to the 'no blur' condition, $t(9) = -21.57, p<0.001$, 95% CI $[-42.09, -34.10]$, Cohen's $d = 7.19$. This effect was more severe for the materials with less scattering in their reflectance functions, which is likely because they were accurately perceived as being glossier in the no blur condition. The steep slopes for the smooth and rough gloss conditions indicate that optical defocus in the foreground rapidly destroyed the cues to gloss generated by the sharp surface ridge, but these large decrements in perceived gloss were not accompanied by large decrements in perceived focus. Together, these findings imply that observers partially misattributed the reduction in image focus in the foreground blur conditions to a change in surface material. Increasing background blur (left side) had a small significant negative effect on perceived gloss for the smooth gloss condition, $t(9) = -4.43, p = 0.002$, 95% CI $[-12.95, -4.20]$, Cohen's $d = 1.48$, but no significant effect for the rough gloss ($t(9) = -0.098, p = 0.924$, 95% CI $[-8.57, 7.85]$]) or matte ($t(9) = 2.52, p = 0.033$, 95% CI $[0.917, 16.940]$]) conditions after the significance criterion was corrected with the Bonferroni method. This is likely due to the loss of gloss cues generated by the sharp specular reflections near the defocused base of the smooth glossy ridge.

Cross-sectional depth profiles of perceived 3D shape constructed from the gauge figure settings are depicted in *Figure 11*. The profiles represent the average across observers after normalizing the mean height of each observer's reconstructed profiles to the overall mean height. Differences in shape between the mean profiles were analyzed with an ANOVA in which each of the twenty probe positions was considered an independent sample; significant main effects of this ANOVA represent systematic changes in shape between pairs of focus conditions. The profiles for both the matte (top) and smooth gloss (bottom) conditions reveal that foreground blur had a significant effect on the perceived 3D shape of the ridge relative to the fully focused conditions, $F(19, 38) = 34.48, p<0.001$ (matte) and $F(19, 38) = 38.84, p<0.001$ (smooth gloss). Background blur had no significant impact on perceived shape for either the matte surface, $F(19, 38) = 1.42, p = 1.78$, or the smooth gloss surface, $F(19, 38) = 0.92, p = 0.569$. The transformations in perceived 3D shape in the foreground blur condition involve a systematic reduction in ridge curvature, height, and position, which suggests that these perceptual distortions were not simply caused by a loss of information. As with perceived gloss, these findings indicate that observers partially misattributed the change in gradient structure induced by foreground blur to a reduction in surface curvature.

Taken together, the results of the three tasks in Experiment 4 reveal that misperceptions of optical defocus are closely coupled with misperceptions of surface properties that also contribute to the smoothness of image gradients. The image features that appear to modulate these misperceptions in our stimuli are the sharp level cut contours that bound the surface ridge. When these contours were defocused by background blur, observers accurately reported a decrease in image focus, but when the ridge was defocused by foreground blur, the largest perceptual changes were instead in material (less gloss) and shape (less curvature). This suggests that the visual system was directly misattributing the smoothness of the image gradients in the

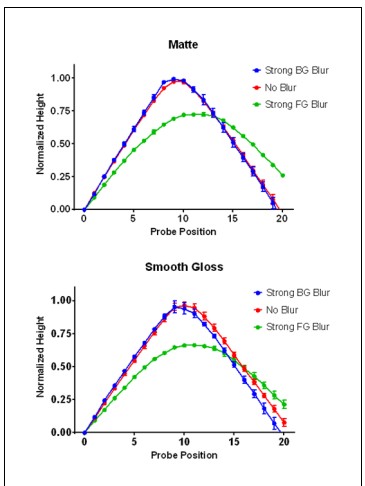

**Figure 11.** Average shape profiles reconstructed from measurements of surface orientation in Experiment 4. The top panel depicts profiles for the matte conditions and the bottom panel depicts profiles for the smooth gloss conditions. The horizontal axis in each plot represents probe location and the vertical axis represents the height of the contour in normalized units of distance relative to the maximum reconstructed height for each observer. The blue, red, and green lines depict the shape profiles for the strong background (BG) blur, no blur, and strong foreground (FG) blur conditions respectively. Error bars represent ± 1 S.E.M. in normalized units.

DOI: https://doi.org/10.7554/eLife.48214.015

foreground blur conditions to the wrong physical sources.

## Discussion

The demonstrations and experiments presented herein were designed to assess how the visual system disentangles the blurred, low-frequency image gradients generated by optical defocus from low-frequency gradients generated by focused shaded surfaces. The experiments were designed to assess the importance of photometric and geometric constraints in resolving this ambiguity. Our results revealed the existence of two types of misperceptions: illusory percepts of optical defocus when none is present (Experiments 1 and 3), and misperceptions of 3D shape and material when defocus is present but not detected (Experiment 4). We found that the perception of optical defocus arose in all shaded stimuli that lacked sharp bounding contours consistent with geometrically-correlated contours such as self-occlusions or level (planar) cuts of the surface. Taken together, our results indicate that this class of bounding contours plays a critical role in the modulating our experience of optical defocus and 3D shape in otherwise ambiguous images of shaded surfaces.

The main theoretical idea that shaped our experiments was that there are specific photogeometric constraints exhibited by the bounding contours of shaded surfaces that play a critical role in identifying low spatial frequency intensity gradients as surface shading, determining the contour's border ownership, and establishing whether the surface is optically focused. We considered two primary constraints. The first was the covariation of contour orientation and shading intensity, which occurs generically for both smooth self-occluding contours and planar cuts (*Figure 2A*). The results of Experiment 1 confirm our informal observations that the rotated level cut mask (*Figure 2B*) not only failed to generate a clear perception of gradient focus, they actually *interfered* with the perception of 3D shape. We attribute this difference to the fact that the contours of level cut masks exhibit a strong orientation-intensity covariation, whereas the contours of the rotated masks do not.

We directly assessed the importance of this constraint in the perception of shading in Experiment 2, where thin shaded 'ribbons' tracing the paths of level cut contours were presented and the intensity-orientation covariation was directly manipulated. Our findings showed that this covariation strongly predicts the perception of 3D shape.

The second constraint we considered involved the direction of curvature of the shading adjacent to a contour (i.e., whether its perceived as convex or concave), and its role in the perceived attachment and focus of the shading gradients. The results of Experiment 1 support our informal observations that concavities appear less focused than convexities. This demonstrates that the intensity-orientation covariation cannot fully explain the perception of focus in images of shaded surfaces, as identical covariation can elicit percepts of both vivid focus and moderate defocus depending on whether the shaded surface is perceived as convex or concave (respectively). The difference in perceived focus may be caused by differences in perceived contour attachment in these two configurations. The shading gradients of the convex surface appear clearly attached to the contours of the level cut, but the same gradients do not appear clearly attached to the contour when the surface appears concave; the level cut contour can appear as the edge of a 'cliff', with the shading appearing at a more distant depth. This suggests that the shading gradients must appear clearly attached to the sharp contour to make full use of the information its sharpness provides about optical focus. This result is also consistent with arguments that the visual system has a bias to interpret abrupt discontinuities in luminance as self-occluding contours rather than sudden changes in surface orientation (*Howard, 1983*; *Liu and Todd, 2004*), which may explain why the contours appear more attached to the shading (and the shading more focused) when the surface appears convex.

Our findings have implications for the existing literature on both the perception of 3D shape from shading and the perception of optical blur. Prior studies of perceived shape have predominantly used globally convex shaded objects as stimuli (e.g. *Fleming et al., 2004*; *Mingolla and Todd, 1986*; *Nefs et al., 2006*). The self-occluding contours that are invariably exhibited by these stimuli may explain why their shading gradients are always perceived as fully focused and vividly three-dimensional. Our data also imply that attempts to 'eliminate' these self-occlusions will only impair perceived 3D shape to the extent that the orientation-intensity covariation exhibited in the image is actually reduced. Artificially cropping a self-occluding contour out of the image, for example, may simply create a new bounding contour that still exhibits enough covariation to induce percepts of focused 3D shading. This may explain why manipulations that rotate shaded terrains (which often

exhit no self-occlusions at all; *Reichel and Todd, 1990*; *Todd and Reichel, 1989*) have been found to negatively affect perceived shape more than cropping the self-occlusions of convex objects (*Fleming et al., 2004*; *Egan and Todd, 2015*). Our results suggest that manipulations involving planar cuts may be a more effective method of investigating the role of contours in surface perception in future work.

The literature on focus perception has predominantly investigated how the severity of perceived blur varies with the spatial frequency and contrast properties of images (*Mather, 1997*; *O'Shea et al., 1997*; *Tadmor and Tolhurst, 1994*). It has been established that relative changes in apparent blur magnitude can provide information about scene depth (*Pentland, 1987*) and that sharp bounding contours can resolve the depth ordering of ambiguous surfaces (*Marshall et al., 1996*), but to our knowledge, no prior studies have examined how the visual system distinguishes gradients produced by optical blur from gradients produced by environmental sources such as shading. Cases of source misattribution, such as those reported here, are likely difficult to produce with the 2D textures and simple contours employed in past work on focus perception. Our data reveal that identical low-frequency shading gradients can be perceived as vividly focused in some contexts and highly blurred in others, which implies that models of focus perception that rely entirely on local gradient features (or even the presence of sharp contours) are not sufficient. Our findings instead suggest that optical focus may be better characterized as a mid-level perceptual category that interacts with the visual system's estimation of other mid-level properties such as contour attachment, 3D surface orientation and curvature, surface reflectance, and scene illumination.

The experiments and demonstrations reported herein have focused on the role of sharp contours that approximate smooth self-occluding rims in providing information about the 3D shape, depth of field, and optical focus of low frequency shading gradients. It seems unlikely, however, that sharp bounding contours are the sole means by which the visual system estimates the optical focus of shaded surfaces. Indeed, we carefully avoided other sources of image contours that could provide information about optical focus, such as the sharp contours generated by either shadows or specular reflections, and only evaluated low-curvature surfaces to avoid generated regions of high spatial frequency shading. Shaded surfaces with regions of high curvature could exhibit enough high spatial frequencies to eliminate percepts of blur, which could explain why some stimuli from prior studies of shading appear focused even in the absence of any contours (e.g. the 'crater' in Figure 7 of *Todd et al., 2014*). Specular reflections generated by low-curvature glossy surfaces also have similar spatial frequency properties to high-curvature shading (*Mooney and Anderson, 2014*). The efficacy of specular reflections in eliminating perceived blur can be experienced directly in the left panel of *Figure 12*. This surface is identical to that depicted in *Figure 6* but was rendered in a natural light field with a specular reflectance component in addition to shading. This image appears as a fully focused, glossy, shaded surface. However, the perception of focus in this image requires that the specular reflections appear linked to the same surface geometry as the shading gradients; if the specular reflections are rotated to arbitrary positions, the shaded surface again appears blurred, and the specular reflections appear as overlaid pigment or a second, independent layer (right panel). Thus, specular reflections must also respect photogeometric constraints that link the specular and diffuse components of reflectance in order to provide information about optical focus, as has been shown previously in the perception of gloss (*Anderson and Kim, 2009*; *Kim et al., 2011*; *Marlow et al., 2012*). The underlying issue in both instances is source attribution: when the relevant image features (image highlights or sharp edges) are attributed to environmental sources (attached specular highlights or attached bounding contours), they simultaneously modulate the perception of environmental properties (gloss or 3D shape) and optical focus. Note that, in general, realistic specular reflections are not optically attached to the surface; when the depth of field is reduced, different regions of specular reflections may only remain sharp when viewed with different focal lengths, and may not be in focus at the same focal length as the attached surface shading. The interactions between specular reflectance, depth of field, and perceived focus are worth consideration in future work.

The demonstrations and experiments presented here have focused on a previously unappreciated computational problem: distinguishing low-frequency structure caused by optical blur from the shading gradients of smooth surfaces. We showed that the visual system exploits specific forms of geometric and photometric covariation to generate percepts of optical focus and vivid surface shading, which arise generically along both smooth self-occlusions and planar cuts. Our results indicate that

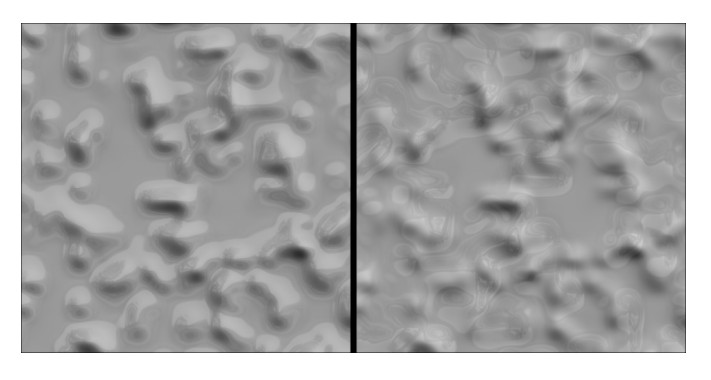

**Figure 12.** The surface used in Experiment 3 has here been rendered with added specular reflections in a natural light field (left). In the right panel, the specular reflections have been rotated by 180 degrees relative to the shading gradients, which breaks their apparent attachment to the surface and reduces the perception of both surface gloss and gradient focus.

DOI: https://doi.org/10.7554/eLife.48214.016

the presence or absence of intensity-orientation correlated contours is a powerful cue to focus: fully focused shaded surfaces can appear blurred in their absence, and actual defocus can be mistaken for transformations in material and shape when correlated contours are nearby. The findings reported herein provide evidence that our perceptions of material, 3D shape, and optical defocus are inherently coupled, which suggests that optical defocus perception does not occur in a completely independent visual pathway to the perception of surface and scene properties. Future work is required to understand the neural processes that exploit these sources of covariation, and the other sources of information that the visual system utilizes to distinguish environmental sources of image structure from optical artifacts induced by the imaging properties of single-chambered eyes.

# Materials and methods

### The orientation-intensity correlation along self-occluding rims and level cuts

The equations for Lambertian shading reveal the correlations between contour orientation and shading intensity that are likely to arise along a level cut. *Equation 1* shows the equation for Lambertian luminance in observer-centric spherical coordinates of surface orientation and illumination direction (*Mamassian, 1993*):

$$L = \max \{ \, 0 \, , \, \, r * i * [\cos(\phi_s)\cos(\phi_i) + \sin(\phi_s)\sin(\phi_i)\cos(\theta_s - \theta_i)] \, \} \tag{1}$$

where $L$ is observed luminance, $r$ is Lambertian surface albedo, and $i$ is the illumination intensity. The sum in square brackets expresses Lambert's cosine law of shading in terms of surface tilt $\theta_s$, surface slant $\phi_s$, illumination azimuth $\theta_i$, and illumination elevation $\phi_i$, all relative to the observer. The subscripts distinguish which pair of spherical coordinates specifies surface orientation ($s$) and which pair specifies the illumination direction ($i$). The maximum function maps negative values of $L$ to zero, which represents surface regions that receive no illumination. If albedo $r$, illumination strength $i$, and illumination direction $(\theta_i, \phi_i)$ are approximately constant across the image, *Equation 1* can be simplified to a function of surface orientation $(\theta_s, \phi_s)$ only:

$$L = \max \{ \, 0 \, , \, \, A\cos(\phi_s) + B\sin(\phi_s)\cos(\theta_s - C) \, \} \tag{2}$$

where $A$ and $B$ are constants determined by surface albedo, illumination strength, and illumination elevation, and $C$ is a constant determined by illumination azimuth. Note that this function is a cosine of surface tilt with phase $C$ whose amplitude and vertical offset may vary with surface slant across

the image. At a self-occluding rim, surface slant approaches 90° (which causes the tilt cosine's vertical offset $A\cos(\phi_s)$ to approach zero and its amplitude $B\sin(\phi_s)$ to approach $B$) and surface tilt is equal to the rim contour's orientation. This further constrains the equation for Lambertian luminance to *Equation 3*:

$$L = \max\{\ 0\ ,\ \ B\cos(\theta_{rim} - C)\ \}\tag{3}$$

where $\theta_{rim}$ is the orientation of the self-occluding rim and $B$ and $C$ are constants that determine the amplitude and phase of the cosine function. Note that the rim's 2D orientation is specified in a full 360° range and not a double-angle 180° range: parallel rim segments bound the surface from opposite sides have opposite orientation (i.e. ± 180°). This equation indicates that at a self-occluding rim, shading luminance decreases as a cosine function of the angular separation between contour orientation and the orientation corresponding to maximal luminance (i.e. where $\theta_{rim} = C = \theta_i$, the illumination azimuth). The phase of the cosine is determined by the illumination azimuth and its amplitude is determined by surface slant, Lambertian albedo, illumination elevation, and illumination intensity. For non-Lambertian diffuse reflectance functions with roughness parameters (e.g. *Oren and Nayar, 1993*), this falloff function will not be an exact cosine, but will at least be monotonic and continuous.

The shading exhibited by surfaces bound by level cut contours is related to contour orientation in a similar way to self-occluding rims. Level cut contour orientation is always equal to surface tilt, but only up to a 180° ambiguity: the adjacent surface could be convex (i.e. the 3D surface normal points out of the shaded region) or concave (i.e. the 3D surface normal points into the shaded region). Slant is unknown along a level cut in principal, but for smooth surfaces, its rate of change along the level cut contour will be constrained and its contribution to shading (the $A\cos(\phi_s)$ and $B\sin(\phi_s)$ terms in *Equation 2*) is consequently likely to be dominated by the contribution of surface tilt (the $\cos(\theta_s - C)$ term in *Equation 2*). The visual system is unlikely to be deterred by any small slant-induced distortions in the shape of the cosine relationship between luminance and contour orientation, which implies that *Equation 3* will still approximately hold for level cuts; the overall correlation between image intensity and orientation across large areas of the image is therefore likely to remain high.

## Experiment 1
### Observers
The exact number of observers recruited for each experiment was dependent on availability; at least ten observers were recruited for all paired comparison and rating tasks described in Experiments 1 to 4, which has been sufficient to detect the effects of material and shape manipulations in our previous psychophysical studies.

Twenty first-year psychology students participated in Experiment one for partial course credit. They all had normal or corrected-to-normal vision and were naïve to the aims of the study.

### Apparatus
Observers were seated approximately 60 cm from a Dell UltraSharp U3014 75.6 cm monitor displaying at a resolution of 2650 × 1600. The display was controlled by a Dell Precision T3600 computer with an Intel Xeon processor running Windows 7 Professional (64-bit). Stimulus presentation and data collection were controlled by OpenGL functions in the Psychophysics Toolbox (version 3.0.10; Brainard, 1997) running in MATLAB (version 2011b; Mathworks, RRID:SCR_001622). Observers were surrounded by a black curtain during the experiment to ensure that the monitor was the primary source of light. Head and eye movements were not restricted. The same apparatus was used for all following experiments.

### Stimuli
A deformed plane was created using the open-source graphics software Blender (version 2.65; Blender Foundation, RRID:SCR_008606). An initial 20 cm square plane was recursively subdivided four times, then deformed by transforming the height of each vertex using a random 'distorted noise' texture generated within Blender, which warps one random noise texture according to the value of another. The contrast of the noise texture was set to 0.25, which ensured that the texture's intensity was globally continuous (i.e. did not clip). A Catmull-Clark subdivision surface modifier with

four iterations was then added to smooth the final surface (*Halstead et al., 1993*). The surface was rendered with Lambertian reflectance (achromatic albedo of $r = 0.3$) under a collimated light source with 90° azimuth (top-down), 45° elevation from the viewing direction, and a strength of 5. The camera was positioned 20 cm away from the center of the deformed plane and set to orthographic view with 1.8 magnification to just crop out the square boundary of the planar surface. The rendered image was saved as a 1024×1024 16-bit TIFF image and tone-mapped to 8-bit grayscale RGB in Adobe Photoshop (Adobe, RRID:SCR_014199). The final image is depicted in *Figure 1*. The procedure used to generate the homogenous gray masks is described in the main text. The final twelve masked stimuli are depicted in *Figure 2—figure supplement 1* ('convex' masks) and *Figure 2—figure supplement 2* ('bistable' masks).

## Procedure

Perceived focus was measured using a paired comparison task. Each of the twelve stimuli was compared with each other stimulus across 66 trials. The stimuli were centered in the left and right halves of the display against a black background. In each trial, observers were instructed to select the image in which the visible gradients appeared most focused. The order of trials and arrangement of stimuli within each trial (left vs. right) were random. Perceived surface curvature sign was measured for the same observers using a three-alternative forced choice task. Each stimulus was presented in the center of the display against a black background above three buttons labeled 'bumps,' 'dents,' and 'neither.' Observers were instructed to click the option in each trial that best represented their overall perception of shape from the visible gradients in the image. They were specifically instructed to inspect the entire image before making their decision. The twelve stimuli were presented once in random order.

## Experiment 2

### Observers

Fifteen observers participated in Experiment 2. One observer was an author (SM). All other observers were recruited from university colleagues, as the study took place outside semester, were naïve to the aims of the studies, and had normal or corrected-to-normal vision.

### Stimuli

Sixteen sets of level cut contours were created by first generating sixteen surfaces similar to the surface in *Figure 1* (using different random noise textures), then intersecting each surface with a fronto-parallel plane. We then used these level cut contours to define the paths of artificial shaded ribbons in MATLAB. The ribbons were centered on the contours and had a uniform width of two pixels. Ribbon orientation was defined in a 360° space by the direction of the 2D normal vector corresponding to the tilt of the terrains used to generate the level cuts (i.e. the normal vectors that point from the gradients into the mask along the level cut mask contours). Ribbon luminance was then defined as a function of ribbon orientation as shown in *Equation 4*:

$$L = 1 - 0.6 * arccos(\langle 0, 1 \rangle \cdot N) \tag{4}$$

where L is shading luminance between zero (black) and one (white) and N is the 2D unit normal vector to the ribbon (which corresponds to 3D surface tilt on the original surface). This equation causes luminance to decrease from 0.8 to 0.2 as a linear function of the angle between ribbon orientation and 90°. The ribbon is brightest where its orientation is equal to 90° and darkest where its orientation is equal to 270°. The background in each image had a luminance value of 0.5, which was the same luminance value exhibited by the exactly vertical segments of the ribbons. These luminance values were scaled to 8-bit achromatic RGB luminance values for display. The resulting shading appears highly similar to the corresponding Lambertian shading on the original rendered surfaces, but varies as a linear function of ribbon orientation instead of obeying Lambert's cosine law. We employed this linear shading approximation to simplify the large volume of correlation analyses performed during the experiment. The difference between these two monotonically decreasing functions is negligible for the purposes of our analysis, as cosine and linear falloff functions have a Pearson correlation coefficient of 0.979. An example of a perfectly correlated ribbon is depicted in the left panel of *Figure 4*.

We systematically weakened the correlation between ribbon orientation and luminance by mixing the linear shading with progressively larger amounts of gradient noise. To accomplish this, we analyzed the Fourier amplitude spectra of the sixteen surfaces used to generate the level cuts for the ribbons and coded a 2D noise generator to generate noise textures that approximately matched the average amplitude spectrum of the shading gradients exhibited by these surfaces. Geometrically identical ribbon paths (two pixels wide) were cut out of these noise textures and mixed with the linearly shaded ribbons in thirteen different proportions. The amount of noise in these mixed ribbons ranged from 0% (which preserves the perfect linear correlation) to 100% (which destroys the linear correlation completely). The intermediate noise values were 15%, 30%, 35%, 40%, 45%, 50%, 55%, 60%, 65%, 70%, and 85%. Note that these values are not spaced at regular intervals, but were instead chosen to generate an approximately uniform distribution of correlation coefficient values. A new random noise texture was generated in real time for each shaded ribbon presented. Example stimuli with 50% and 100% noise are shown in *Figure 4*.

We measured the global correlation between contour orientation and shading luminance in each ribbon image presented to observers. This was accomplished in several steps for each image. First, we used the in-built *edge* function in MATLAB to extract the contours from the binary level cut mask. We then calculated discrete approximations of contour orientation in 9 × 9 blocks of pixels using the *regionprops* function, but the resulting orientation values only range from 0° to 180° and do not distinguish between parallel contour segments with opposing normal vectors (i.e. on opposite sides of the mask). We rectified this using a custom program that tests which side of the contour corresponds to the mask and which corresponds to the visible gradients at each pixel along the contour. Contour segments that were detected as bounding the mask from above had their orientation values translated into a range from 180° (a right mask edge) to 360° (a left mask edge). These orientation values match the orientation of 2D normal vectors pointing outward from the gradients into the mask (which would in turn correspond to surface tilt at a convex level cut or self-occluding rim). The code for ribbon generation and correlation computation is available on GitHub (*Mooney, 2019*; copy archived at https://github.com/elifesciences-publications/contour-covariation).

We calculated sixteen orientation- correlations for each image using sixteen different potential values for the illumination azimuth, which determines which contour orientation corresponds to the brightest shading. The azimuth values increased from 0° to 337.5° in increments of 22.5°. These multiple correlations were measured to account for the possibility that the ribbon noise would induce incidental correlations consistent with illumination directions other than the original top-down direction. Each correlation was calculated by converting the contour orientation values into angular separation values from the azimuth, then computing the Pearson correlation coefficient between angular separation and image intensity at each pixel along the ribbon. The final correlation value $\rho$ for that stimulus was set to the maximum correlation detected across all sixteen potential illumination azimuth values. The correlations were derived in real time throughout the task due to the random noise used to generate each stimulus. Correlations for the example stimuli depicted in *Figure 4* are shown in the bottom-right corner of each image. Note that the shaded ribbons with no gradient noise exhibit almost perfect correlations as designed. All correlation coefficients greater than 0.2 were consistent with 90° illumination azimuth, which indicates that there were no strong correlations consistent with arbitrary illumination directions in the experiments.

## Procedure

Perceived 3D shape strength was measured using a paired comparison task. Observers were shown all possible pairs of the thirteen ribbon noise values in random order. The shaded ribbon paths displayed in each trial were randomly chosen from the sixteen possible sets of contours and were mixed with new randomly generated noise textures in every trial. Observers were instructed to select the image that appeared most vividly three-dimensional overall and were told to fully inspect both images before deciding. The orientation-intensity correlation for each stimulus was derived in real time during the experiment for later analysis.

## Experiment 3

### Observers

Ten observers participated in Experiment 3. One observer was the author (SM). All other observers were recruited from university colleagues, were naïve to the experimental aims, and had normal or corrected-to-normal vision.

### Stimuli

The stimulus geometry was created in the same way as Experiment 1, but the noise texture used to deform the surface had a contrast value of 1. This clipped the noise intensity to black below a level of 0.5, which restrained the height of the deformed surface to a minimum value (i.e. the concave parts of the surface in *Figure 1* became flat). The same subdivision modifier was then applied to the surface to smooth over the abrupt discontinuities in surface orientation created by the clipped noise, and the material, illumination, and camera parameters were otherwise identical. A level cut condition was then re-created from the smoothed surface by truncating it with a flat frontoparallel plane positioned 0.1 cm in front of the flat planar regions. The resulting combined surface exhibits sharp level cut contours along the paths where the surfaces intersected, which are similar to the contours in Experiment 1.

Five optical defocus conditions were applied to both the smoothed and level cut geometry conditions by manipulating virtual camera parameters. In the 'no blur' condition, no defocus was used. The four conditions with blur were created by combining two different focal lengths with two lens aperture values. In the two 'background blur' conditions, focal length was set to the average distance of the peaks of the five highest bumps (19.3 cm). In the 'foreground blur' conditions, focal length was set to the distance of the intersecting plane (19.9 cm). The weak and strong blur conditions for each focal length were defined by setting the F-stop value to 2.8 and 1.4, respectively. The stimuli were rendered to 1024 × 1024 16 bit TIFF images, collated, and simultaneously tone-mapped to 8-bit RGB in Adobe Photoshop. The final ten stimuli are depicted in *Figure 6*. The top row depicts the smoothed surface with no intersecting plane and the bottom row depicts the level cut surface with the intersecting plane. From left to right, the columns depict the strong background blur, weak background blur, no blur, weak foreground blur, and strong foreground blur conditions.

### Procedure

Perceived focus was measured using a paired comparison task identical to the task used in Experiment 1, but due to the lower number of stimuli, the full block of 45 pairs was shown twice in random order to each observer.

## Experiment 4

### Observers

Twenty first-year psychology students participated in the focus and gloss tasks of Experiment four for partial course credit. They all had normal or corrected-to-normal vision and were naïve to the aims of the study. Five psychophysically experienced observers participated in the shape task, including an author (SM). This number of observers has been sufficient to detect the effects of surface manipulations on the gauge figure task, which involves a very large number of trials. All observers other than the author were naïve to the aims of the study.

### Stimuli

The deformed planar surface was created in a similar way to Experiment 3, but the surface was not subdivided after deformation with the high-contrast noise texture. This creates sharp level-cut discontinuities in shape that separate smooth bumps from a planar background, and was done to effectively include the intersecting plane as part of the rendered scene rather than adding it to the image as a mask. Three different reflectance functions were used to create low-roughness specular ('smooth gloss'), moderate-roughness specular ('rough gloss'), and shading-only ('matte') conditions. All three reflectance functions had a diffuse Lambertian component with 95% strength and achromatic albedo $r = 0.3$. The smooth gloss and rough gloss conditions had additional specular

reflectance components defined by the Beckmann BRDF with 5% strength and roughness coefficients $k = 0.025$ (smooth gloss) and $k = 0.1$(rough gloss).

We created five different blur conditions by simulating different types of optical defocus in Blender. This was accomplished by changing the properties of the virtual camera, which was positioned at 20 cm from the planar base of the bumpy surface. The 'no blur' condition had no optical defocus. In the 'foreground blur' and 'background blur' conditions, the camera's focal length was set to the distance between the camera and the planar base of the surface (20 cm) or the closest point on the main visible ridge in the center-left part of the image (18.5 cm), respectively. The weak and strong versions of each blur condition were generated by setting the virtual camera's aperture to F-stop values of 2.8 and 1.4, respectively. Camera zoom was set to 150 in all conditions to just crop out the square boundary of the deformed plane. The stimuli were illuminated using a natural light field to test whether misperceptions of optical blur occur in more realistic viewing conditions, which are particularly important for the appearance of glossy materials (*Fleming et al., 2003*; *Pellacini et al., 2000*). The light field used was 'Meadow Trail', which we have used previously (*Mooney and Anderson, 2014*). The images were rendered to 1024 × 1024 16 bit TIFF images in Blender, then simultaneously tone-mapped to 8-bit RGB in Adobe Photoshop. The final fifteen stimuli are depicted in *Figure 8*. The top, middle, and bottom rows show the matte, rough gloss, and smooth gloss materials, respectively. From left to right, the columns depict the strong background blur, weak background blur, no blur, weak foreground blur, and strong foreground blur conditions.

### Procedure

Perceived focus and gloss were measured using paired comparison tasks in the same way as Experiment 1. Perceived 3D shape was measured for six of the stimuli: the smooth gloss and matte surfaces with no blur, strong background blur, and strong foreground blur (outlined in red in *Figure 8*). We employed the gauge figure task to measure shape (see *Koenderink et al., 1992*). Observers adjusted gauge figure probes at twenty points in a horizontal line along the central ridge in the bottom-left quadrant of each image. In each trial, observers adjusted the gauge figure until its disc appeared to lie in the tangent plane to the surface and its rod matched the surface normal. Observers completed four repeats of each of the six images. The images were presented in pseudo-random order to ensure no image was seen twice in a row and the twenty probes for each image were presented in random order. No time limit was given, and observers took approximately forty minutes to complete the experiment.

## Additional information

### Funding

| Funder | Grant reference number | Author |
|---|---|---|
| Commonwealth of Australia | Australian Postgraduate Award | Scott WJ Mooney |

The funders had no role in study design, data collection and interpretation, or the decision to submit the work for publication.

### Author contributions

Scott WJ Mooney, Conceptualization, Software, Formal analysis, Investigation, Methodology, Writing—original draft, Writing—review and editing; Phillip J Marlow, Conceptualization, Writing—review and editing; Barton L Anderson, Supervision, Investigation, Writing—review and editing

### Author ORCIDs

Scott WJ Mooney (iD) https://orcid.org/0000-0002-0094-7638

### Ethics

Human subjects: Informed consent and consent to publish was obtained from each participant in accordance with experimental protocol 2012/2759 approved by the Human Research Ethics Committee (HREC) at the University of Sydney.

## Decision letter and Author response

Decision letter https://doi.org/10.7554/eLife.48214.019
Author response https://doi.org/10.7554/eLife.48214.020

---

## Additional files

### Supplementary files

• Transparent reporting form
DOI: https://doi.org/10.7554/eLife.48214.017

### Data availability

All data generated or analysed during this study are included in the manuscript and supporting files. The code for ribbon generation and correlation computation is available on GitHub (https://github.com/swjmooney/contour-covariation; copy archived at https://github.com/elifesciences-publications/contour-covariation).

---

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
