## [Decision Letter]

Thank you for submitting your article "The perception and misperception of optical defocus, shading, and shape" for consideration by *eLife*. Your article has been reviewed by three peer reviewers, including Roland W Fleming as the guest Reviewing Editor and Reviewer #1, and the evaluation has been overseen by Joshua Gold as the Senior Editor. The following individual involved in review of your submission has also agreed to reveal their identity: James Todd (Reviewer #2).

The reviewers have discussed the reviews with one another and the Reviewing Editor has drafted this decision to help you prepare a revised submission.

As you will see below, the reviewers were all quite positive. We therefore are providing all of the comments from the individual reviewers in full so you can decide how you wish to respond to their particular suggestions.

*Reviewer #1:*

This is a highly original and insightful contribution, which shows for the first time how mid-level perceptual organization processes affect the apparent sharpness (focus) of images. The main, and highly surprising, finding is that identical smooth intensity gradients can appear either blurry or sharp depending on the extent to which sharp contours that encompass them appear to be 'owned by' the image region containing the smooth gradient. When the sharp contours appear to be the boundary of the smooth image region, the gradients appear like shading seen in focus. In contrast, in the absence of such contours, or when they appear to belong to an overlying occluder, the smooth gradients are seen to be out of focus, as if the visual system attributes the smoothness of the gradients to optical blur, rather than shading.

To my knowledge, no previous studies have considered this causal attribution ambiguity, and the findings have implications for our understanding of 3D shape perception and image interpretation more generally. The study elegantly combines photo-geometric insights derived from considering the generative processes of shading, with phenomenology and behavioural experiments to map out the range of conditions that modify the perception of image sharpness based on border ownership.

Having said that, there are a couple of aspects of the manuscript that could be improved.

My first major suggestion is to remove Experiment 2. It is not that the experiment is any way flawed, it's just that the motivation for the experiment and its connection to the other experiments is not sufficiently clear in the manuscript. In short, it's not clear what it adds, as none of the main arguments in the article require the evidence provided by Experiment 2. Moreover, of all the experiments, it is the one with the least surprising result in light of what we already know about shape from shading, including the authors' own work. Removing it would help streamline what is a rather long manuscript.

My other main comment is that I suspect that the key driver of apparent sharpness is the extent to which sharp features appear to 'belong to' the surface. This is indeed what the authors say (e.g. Discussion, fourth paragraph), but they emphasise the 'photogeometric constraint' of intensity-orientation correlation along the contour as the main driver of the effect. I agree that this affects perceived sharpness because it mediates border ownership. But I think it is probably only one of several ways of getting the sharpness to 'adhere to' the smooth gradients (as indeed the authors note in the seventh paragraph of the Discussion). For example, sharp features that look like surface markings should also make the gradients appear in focus. It would be interesting, for example to make a stereoscopic version of Figure 1 and add a few dots on top. Placing the dots in a plane floating above the surface should leave it looking blurry. But placing the dots stereoscopically on the relief of the surface should make it look sharp because they 'belong to' the surface. If my intuition is correct, it would generalize the results the authors present. If I'm wrong, that would also be interesting as it would demonstrate a truly privileged status for boundary ownership and the intensity-orientation and convexity constraints.

Overall, however, this is a compelling and important contribution.

*Reviewer #2:*

This is an outstanding manuscript. The methodology of the work is highly creative, and the paper is clearly written. Because this work makes an important contribution to the field of perception, I strongly recommend that it be accepted for publication.

*Reviewer #3:*

This is a scholarly piece of work which presents very clear and compelling results which will be of great interest to the, albeit rather small, contingent of people interested in shape from shaping, the perceptions of surface material and blur. The paper presents a novel approach and is certainly worthy of publication. It is very well written. Very unusually for me, I do not have much to say by way of critique – the paper is very good in my view. My only real concern is that, if I read the paper correctly, each experiment relied on small number of rendered surfaces (perhaps even just one) and a relatively small number of stimuli (perhaps just one per conditions) presented a small number of times to a relatively large number of people. It is possible then than some effects rely on or are enhanced by very specific image features that happened to occur in these stimuli. It would have been better if multiple surfaces had been used in each experiment.

---

## [Author Response]

Reviewer #1:

[…] There are a couple of aspects of the manuscript that could be improved.My first major suggestion is to remove Experiment 2. It is not that the experiment is any way flawed, it's just that the motivation for the experiment and its connection to the other experiments is not sufficiently clear in the manuscript. In short, it's not clear what it adds, as none of the main arguments in the article require the evidence provided by Experiment 2. Moreover, of all the experiments, it is the one with the least surprising result in light of what we already know about shape from shading, including the authors' own work. Removing it would help streamline what is a rather long manuscript.

We agree that Experiment 2 did not directly address the issue of optical defocus; its purpose was to directly test the idea that the photogeometric covariation of intensity and contour orientation was a sufficient cue that the visual system used to identify shading, which in turn supports our explanation for the role of these contours in Experiment 1. Although we have done other manipulations in a previous paper to assess this as well, this is, in our minds, a more direct test. We have clarified its relevance to questions of perceived focus in our revision:

“These findings do not directly address the issue of perceived defocus, but do support our hypothesis that the photo-geometric behavior occurring at the very edge of covarying contours (such as the level cut contours in Figure 2A) is sufficient to generate the vivid impressions of contour attachment observed in the ‘convex’ conditions of Experiment 1.”

It is only a short component of the paper, so we feel that there is no good reason to omit it since it adds information demonstrating the importance of this covariation in identifying shading.

My other main comment is that I suspect that the key driver of apparent sharpness is the extent to which sharp features appear to 'belong to' the surface. This is indeed what the authors say (e.g. Discussion, fourth paragraph), but they emphasise the 'photogeometric constraint' of intensity-orientation correlation along the contour as the main driver of the effect. I agree that this affects perceived sharpness because it mediates border ownership. But I think it is probably only one of several ways of getting the sharpness to 'adhere to' the smooth gradients (as indeed the authors note in the seventh paragraph of the Discussion). For example, sharp features that look like surface markings should also make the gradients appear in focus. It would be interesting, for example to make a stereoscopic version of Figure 1 and add a few dots on top. Placing the dots in a plane floating above the surface should leave it looking blurry. But placing the dots stereoscopically on the relief of the surface should make it look sharp because they 'belong to' the surface. If my intuition is correct, it would generalize the results the authors present. If I'm wrong, that would also be interesting as it would demonstrate a truly privileged status for boundary ownership and the intensity-orientation and convexity constraints.

This was our intuition as well, but it turns out that generating surface textures that appear to ‘belong to’ the smooth shading is not as simple as one might expect. See our 2018 VSS presentation:

Mooney, S.W.J. and Anderson, B.L. (2018). Illusory transparency and optical blur induced by single shaded surfaces. Journal of Vision 18(10), 889-889.

Our original thought was to simply add some high frequency texture (fine grained 3D bumps) and render the surface. We thought this should obviously appear in focus, but it doesn’t; the high frequency component does not appear attached to the surface, but instead appears as a fine-grained noise layer through which the surface is viewed, which still appears blurry. This occurs even in stereo (to a lesser extent), as though depth cues from the apparent defocus of the contour-less shaded surface are actively counteracting the stereo depth cues. This is a paper we are currently in the process of writing up. We have not raised it here because it would require going into the details of the results; it will be discussed in the texture follow-up to this work.

Overall, however, this is a compelling and important contribution.

Reviewer #3:

[…] Very unusually for me, I do not have much to say by way of critique – the paper is very good in my view. My only real concern is that, if I read the paper correctly, each experiment relied on small number of rendered surfaces (perhaps even just one) and a relatively small number of stimuli (perhaps just one per conditions) presented a small number of times to a relatively large number of people. It is possible then than some effects rely on or are enhanced by very specific image features that happened to occur in these stimuli. It would have been better if multiple surfaces had been used in each experiment.

This is a fair point. We have generated and examined many variations of the smooth surface geometries included in the paper, and have found that the effects we describe are very general as long as the surface relief and/or illumination elevation are not large enough to induce cast shadows in the image (which introduces additional complex cues to shape and focus). As the exact geometry was otherwise not relevant, we chose to include a greater number of manipulations of other stimulus parameters such as mask type, material, and defocus severity rather than additional geometries.